# Potency of gastrointestinal colonization and virulence of *Candida auris* in a murine endogenous candidiasis

Masahiro Abe[1], Harutaka Katano[2], Minoru Nagi[1], Yoshitsugu Higashi[1], Yuko Sato[2], Ken Kikuchi[3], Hideki Hasegawa[2,4], Yoshitsugu Miyazaki[1]*

1 Department of Chemotherapy and Mycoses, National Institute of Infectious Diseases, Tokyo, Japan, 2 Department of Pathology, National Institute of Infectious Diseases, Tokyo, Japan, 3 Department of Infectious Diseases, Tokyo Women's Medical University, Tokyo, Japan, 4 Influenza Virus Research Center, National Institute of Infectious Diseases, Tokyo, Japan

* ym46@niid.go.jp

**Data Availability Statement:** All relevant data are within the manuscript and its Supporting information files.

## Abstract

### Background

*Candida auris* infections have recently emerged worldwide, and this species is highly capable of colonization and is associated with high levels of mortality. However, strain-dependent differences in colonization capabilities and virulence have not yet been reported.

### Objectives

In the present study, we aimed to clarify the differences between clinically isolated invasive and non-invasive strains of *C. auris*.

### Methods

We evaluated colonization, dissemination, and survival rates in wild C57BL/6J mice inoculated with invasive or non-invasive strains of *C. auris* under cortisone acetate immunosuppression, comparing with those of *Candida albicans* and *Candida glabrata* infections. We also evaluated the potency of biofilm formation.

### Results

Stool fungal burdens were significantly higher in mice inoculated with the invasive strains than in those infected with the non-invasive strain. Along with intestinal colonization, liver and kidney fungal burdens were also significantly higher in mice inoculated with the invasive strains. In addition, histopathological findings revealed greater dissemination and colonization of the invasive strains. Regarding biofilm-forming capability, the invasive strain of *C. auris* exhibited a significantly higher capacity of producing biofilms. Moreover, inoculation with the invasive strains resulted in significantly greater loss of body weight than that noted following infection with the non-invasive strain.

**Funding:** This work was supported in part by the Research Program on Emerging and Re-emerging Infectious Diseases of the Japan Agency for Medical Research and Development (AMED) under Grant Number (JP19fk0108045, JP19fk0108049, JP19fk0108094, and JP19fk0108104), and by JSPS KAKENHI Grant Number JP17K10040. JP19fk0108045, JP19fk0108049, JP19fk0108094, and JP17K10040 were given to YM. JP19fk0108104 was given to HK and HH. URL of AMED is https://www.amed.go.jp/en/index.html URL of JSPS KAKENHI is https://www.jsps.go.jp/english/index.html The funders had no role in study design, data collection and analysis, decision to publish, or preparation of the manuscript.

**Competing interests:** The authors have declared that no competing interests exist.

## Conclusions

Invasive strains showed higher colonization capability and rates of dissemination from gastrointestinal tracts under cortisone acetate immunosuppression than non-invasive strains, although the mortality rates caused by *C. auris* were lower than those caused by *C. albicans*.

## Introduction

*Candida* species are common causative agents of bloodstream infections in hospitals, leading to high levels of mortality and morbidity [1,2]. There are two routes that *Candida* species follow for infecting the bloodstream as follows: exogenous or endogenous [3–6]. Endogenous bloodstream infections are considered to be caused by the invasion of *Candida* species colonizing in mucosa, on the other hands, exogenous infections are often caused as catheter-related bloodstream infection [6–8]. In particular, the previous report describes gastrointestinal tract is one of the primary areas of *Candida* species colonization and these species can invade and disseminate from this location, leading to bloodstream infection under specific conditions [6].

*Candida auris* was initially isolated and reported in a Japanese patient with chronic otitis media [9]. After the first isolation, *C. auris* infections were reported worldwide in places such as Korea, India, the United States, South America, and South Africa [10–16]. In several reports, *C. auris* bloodstream infections were associated with high mortality and morbidity, which indicated the high virulence of clinically isolated *C. auris* strains. In mouse experiments, *C. auris* strains clinically isolated from bloodstream infections were reportedly highly virulent, leading to severe infections and high mortality rates [17–19]. In these experiments, systemic candidiasis was caused by intravenous injection mimicking exogenous candidiasis; however, to the best of our knowledge, no model of endogenous candidiasis dissemination from the gastrointestinal tract caused by *C. auris* has yet been reported. *C. auris* was reported to colonize on mucocutaneous surfaces, especially skins, and healthcare environments, leading to bloodstream infections via catheters, although some articles have reported *C. auris* gastrointestinal colonization and the possibilities of dissemination [20–22]. Therefore, studies of capacity for colonization and dissemination from gastrointestinal tracts are necessary. In addition, 4 clades of *C. auris*; East Asian, South Asian, South America, and African clade; have been reported, and all *C. auris* isolates in Japan belonging to the East Asian clade were obtained from the patients with chronic otitis media [23]. However, *C. auris* strains found in other countries, belonging to the South Asian, South American, or African clade, were primarily isolated from bloodstream infections. Hence, differences in virulence and colonization capabilities between *C. auris* strains in Japan (non-invasive strains) and those from other countries (invasive strains) may exist, but research into these strain-specific differences is scarce.

In this report, we compared the capability for colonization and dissemination from gastrointestinal tracts between a non-invasive strain of *C. auris* isolated in Japan and invasive strains isolated in other countries using an endogenous candidiasis mouse model under cortisone acetate immunosuppression. We also evaluated the pathogenicity of *C. auris* strains compared with other pathogenic *Candida* species using survival analysis.

## Material and methods

### Mice

Male C57BL/6J mice, 6–7 weeks old, were purchased from Japan SLC, Inc. (Shizuoka, Japan) and maintained under specific pathogen-free conditions at the National Institute of Infectious Diseases in Japan. All experiments were reviewed and approved by the Animal Care and Use Committee of the National Institute of Infectious Diseases. All staffs including this investigation were sufficiently educated in animal care and handling before this procedure. Protocols were designed for minimizing animal suffering and limit the numbers used in experiments (Approval numbers:117139 and 119047).

### Yeast strains and infection models

Two clinically isolated non-invasive strains of *C. auris* (JCM 15448: the first strain isolated in Japan [7], and TWCC 58362) and invasive strains obtained from the Center of Disease Control and Prevention AR Isolate Bank collection of *C. auris* isolates (AR0382, AR0383, AR0385, and AR0390) were used in this study (Table 1). Reference strains of *Candida albicans* (SC5314) and *Candida glabrata* (CBS 138) were used in the survival analysis.

For the *Candida* species colonization and dissemination mouse model, mice were administered with a cocktail of antibiotics in their drinking water for more than 1 week before inoculation with *C. auris*. The cocktail consisted of vancomycin (45 mg/L), gentamicin (35 mg/L), kanamycin (400 mg/L), metronidazole (215 mg/L), and colistin (850 U/L), as previously reported for the mouse model of *Clostridioides difficile* infection [24]. *C. albicans* and *C. auris* (AR0382, AR0383, AR 0385, and AR0390) were grown at 30˚C and *C. glabrata* at 37˚C for 2 days on yeast extract peptone dextrose (YPD) agar. *C. auris* (JCM 15448 and TWCC 58262) was grown at 30˚C for 3 days on YPD agar. Next, *C. albicans* and *C. auris* were grown at 30˚C and *C. glabrata* was grown at 37˚C in YPD broth for 18–24 h. Following incubation, the yeasts were collected, washed, and resuspended in sterile normal saline at approximately $5.0 \times 10^8$–$1.0 \times 10^9$ colony-forming units (CFU)/mL. Mice were inoculated with 100 μL (approximately $5.0 \times 10^7$–$1.0 \times 10^8$ CFU/mouse) via a sterile feeding needle for instigating the colonization of the *Candida* species. One day previously, on the day, and 1 day after inoculation with the *Candida* species, cortisone acetate (225 mg/kg) was subcutaneously injected into each mouse, as previously reported for the mouse model of oropharyngeal candidiasis with modifications [25].

### Evaluation of organ dissemination and gastrointestinal colonization

In these experiments, mice were divided into six groups as follows: *C. auris* JCM 15448, *C. auris* TWCC 58362, *C. auris* AR0382, *C. auris* AR0383, *C. auris* AR0385, or *C. auris* AR0390 inoculated groups. These mice were euthanatized 7 or 14 days after inoculation, followed by

**Table 1. *Candida auris* strains used in the study.**

| Strain | Clade | Origin |
|---|---|---|
| JCM 15448 | East Asian | Chronic otitis media |
| TWCC 58362 | East Asian | Chronic otitis media |
| AR0382 | South Asian | Bloodstream infection |
| AR0383 | African | Bloodstream infection |
| AR0385 | South American | Bloodstream infection |
| AR0390 | South Asian | Bloodstream infection |

the aseptic collection of liver and kidneys. Prior to euthanasia, fresh stools from each mouse were collected for evaluating gastrointestinal colonization. The liver, kidneys, and stools were then homogenized in sterile PBS, and the homogenates were serially diluted. Liver and stool homogenates were plated on YPD agar with antibiotics (penicillin/streptomycin, to inhibit bacterial growth), and kidney homogenate was plated on YPD agar, both at 100 μL per plate. The fungal burden was determined by counting CFUs after 24–48 h of incubation at 30˚C.

### Histopathological analysis of organs from infected mice

For histopathological analysis, liver, kidneys, and colons were removed 14 days after *C. auris* inoculation and fixed in 10% neutral buffered formalin, dehydrated with ethanol, and embedded in paraffin following standard procedures. Tissue sections (4 μm) were mounted onto glass slides (Matsunami Glass, Osaka, Japan) and stained with hematoxylin and eosin or Grocott methenamine silver (GMS). After staining, histological examinations were performed using light microscopy.

### Biofilm-forming assay

In this assay, invasive strains of *C. auris* (AR0382, AR0383, AR0385, or AR0390) were grown for 2 days and non-invasive strains of *C. auris* (JCM 15448 and TWCC 58362) for 3 days at 30˚C on YPD agar. Each *C. auris* strain was then grown at 30˚C in YPD broth for 18–24 h. After incubation, yeasts were collected, washed with sterile PBS, resuspended in YPD broth, and diluted to $1.0 \times 10^8$ cells/mL. First, 100 μL of this inoculum was added to each well of a 96-well plate and incubated for 4 h to allow the cells to adhere to the plastic surface, as previously described, with minor modifications [26,27]. In the negative control wells, 100 μL of YPD broth was added. After 4 hours of adhesion, the supernatant of each well was discarded to remove planktonic *C. auris* cells. After that, 100 μL of YPD broth were added to each well. The plate was then incubated for 24 h to allow *C. auris* to form a biofilm. After 24 h of incubation, the supernatant in each well was carefully removed and 150 μL of methanol were added to fix the biofilm. After methanol fixation, each well was washed three times with PBS and stained with 0.1% crystal violet for 60 min. After crystal violet staining, each well was washed several times with PBS. Lastly, the crystal violet bound to the *C. auris* biofilms was dissolved with 200 μL of methanol and the optimal density at 595 nm was measured using a microtiter plate reader.

### Survival analysis

In this experiment, mice were divided into four groups as follows: *C. auris* JCM 15448, *C. auris* AR0382, *C. albicans* SC5314, or *C. glabrata* CBS 138 inoculated groups and survival rates were recorded for 4 weeks after inoculation. Mice were monitored at least once a day and the body weights of these inoculated mice were also monitored. Any mice that appeared to be unable to maintain standing position, loss of appetite and drinking, hunched position, or moribund (reduced activity, tachypnea, and poorly responsive to external stimuli) were humanely euthanized as soon as possible by the inhalation of carbon dioxide. All mice remaining at the end of the survival analysis were also euthanized by the inhalation of carbon dioxide. In total, 52 mice were used and divided into 4 groups in this analysis.

### Statistical analysis

Continuous variables were compared among groups by the Kruskal-Wallis test and post hoc Dunn's test for pair-wise comparisons. In the survival analysis, groups were compared using

the log-rank (Mantel–Cox) test. Body weight changes during the survival analysis were compared using two-way ANOVA followed by post hoc Tukey's multiple comparisons test. A *P* value of < 0.05 was considered significant for all tests. Statistical analyses were performed using GraphPad Prism, version 7 (GraphPad Software, La Jolla, CA, USA).

# Results

## Potency of gastrointestinal colonization of *C. auris*

We intragastrically inoculated each mouse with non-invasive strains of *C. auris* JCM 15448 or *C. auris* TWCC 58362, or invasive strains AR0382, AR0383, AR0385, or AR0390. After inoculation, we evaluated the gastrointestinal colonization of each group by calculating stool CFUs. The fungal burden in the gastrointestinal tract 7 days after inoculation was significantly higher in mice infected with invasive strains, especially AR0382 or AR0390, compared with those infected with non-invasive strains (Fig 1A). Similarly, 14 days after inoculation, the stool fungal burden in each group inoculated with AR0382 or AR0390 was also significantly higher compared with that in the group infected with the non-invasive strain (Fig 1B). No significant differences were noted between mice inoculated with the invasive strains. Based on the above results, we histopathologically observed gastrointestinal colonization of murine colons 14 days after inoculation with non-invasive strains of *C. auris* JCM 15448, and invasive strains AR0382 or AR0390. In the colon, GMS staining demonstrated abundant levels of yeasts in the stools and on the surfaces of intestinal mucosa in mice inoculated with AR0382 or AR0390, but sparse levels in mice inoculated with JCM 15448 (Fig 2A–2C). Neither yeast invasion of the mucosa nor any filamentous forms were observed in any mice. These results demonstrated that each *C. auris* strain could colonize the gastrointestinal tracts following treatment of the mice with cortisone acetate and antibiotics. In addition, the invasive strains colonized the murine gastrointestinal tract more effectively than the non-invasive strain.

## Dissemination of *C. auris* from gastrointestinal tracts to organs

In addition to measuring gastrointestinal colonization, we also evaluated the dissemination of the yeasts from the gastrointestinal tracts. Seven days after inoculation, the fungal burdens of the liver and kidneys were significantly higher in mice infected with the invasive strains, especially AR0382 or AR0390, compared with those in mice infected with the non-invasive strains (Fig 3A and 3B). Moreover, there were no differences of organ disseminations among invasive strains. At 14 days after inoculation, the fungal burdens of the liver and kidneys of mice inoculated with the invasive strains were also significantly higher than those with the non-invasive strains (Fig 3C and 3D). No significant differences were noted between mice inoculated with the invasive strains. In some mice infected by the non-invasive strains, dissemination into liver or kidneys was found 7 or 14 days after inoculation; however, the dissemination rates in mice infected with this non-invasive strain were comparably small. In addition, on the basis of the results obtained from the dissemination experiments, we histopathologically evaluated the liver and kidneys of mice 14 days after inoculation with the invasive strains AR0382 or AR0390. The histological analyses revealed that round yeasts without hyphae formed a fungal mass in the organs of mice infected with AR0382 or AR0390 (Fig 4A and 4B). In the liver, focal necrosis with fungi and neutrophil infiltration was observed, although little inflammation was observed in the portal area. In the kidney, yeasts were detected in renal tubules with inflammatory cell infiltration. However, no yeast cells were detected in the liver and kidneys of mice infected with JCM 15448 (S1 Fig). Taken together, these results indicated that the invasive strains isolated from the bloodstream infections were more capable of colonizing and

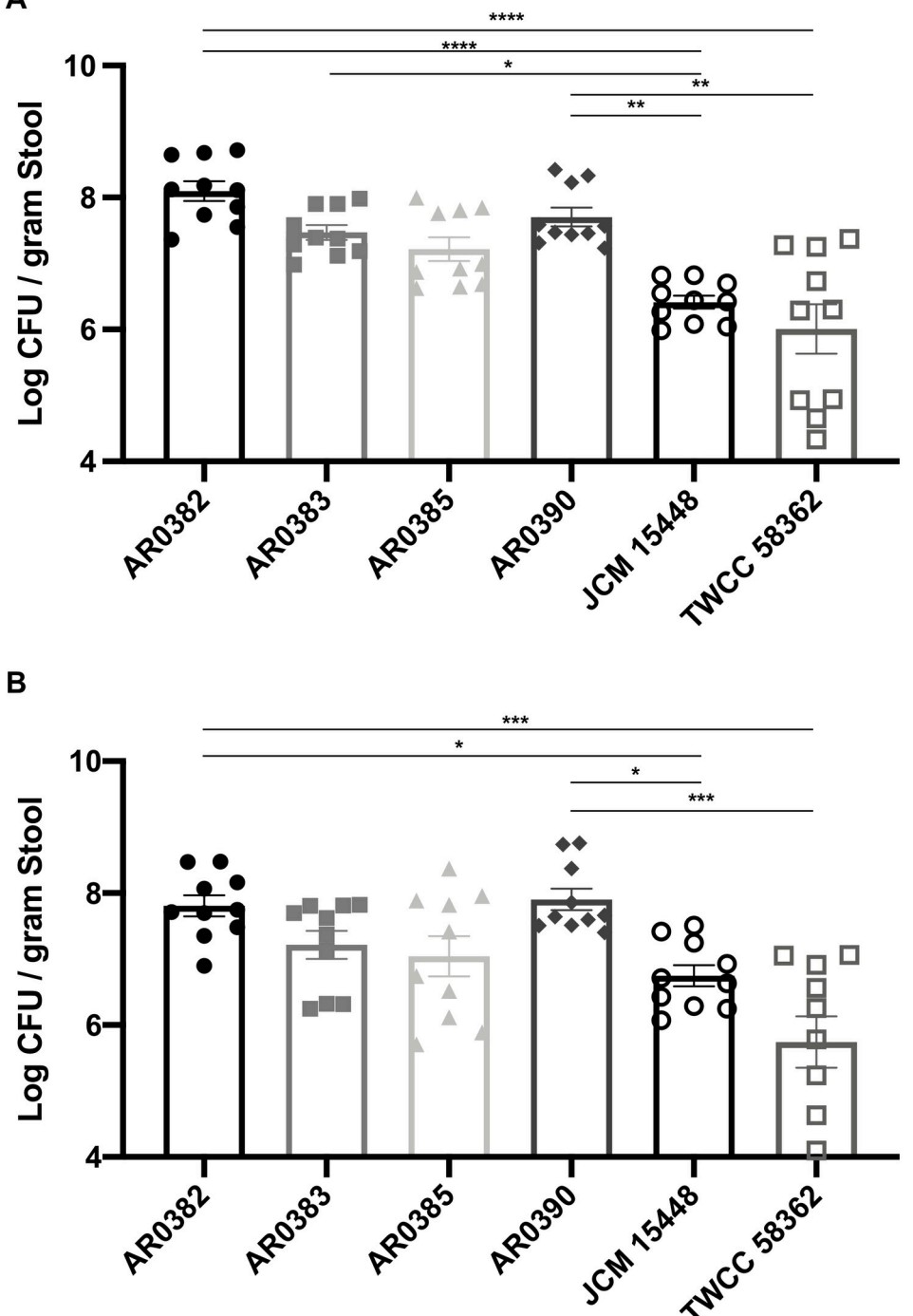

**Fig 1. Stool fungal burden was significantly higher in mice infected by invasive strains of *Candida auris* strains than in those infected with the non-invasive strains.** (A, B) The fungal burdens in stools collected 7 days (A) and 14 days (B) after inoculation of invasive or non-invasive *C. auris* strains. Stool fungal burdens are shown as the Log CFU/ g stool. All results are expressed as mean ± standard error of the mean from six independent experiments with a total of 10 stool samples per group. *** $P < 0.001$, **** $P < 0.0001$.

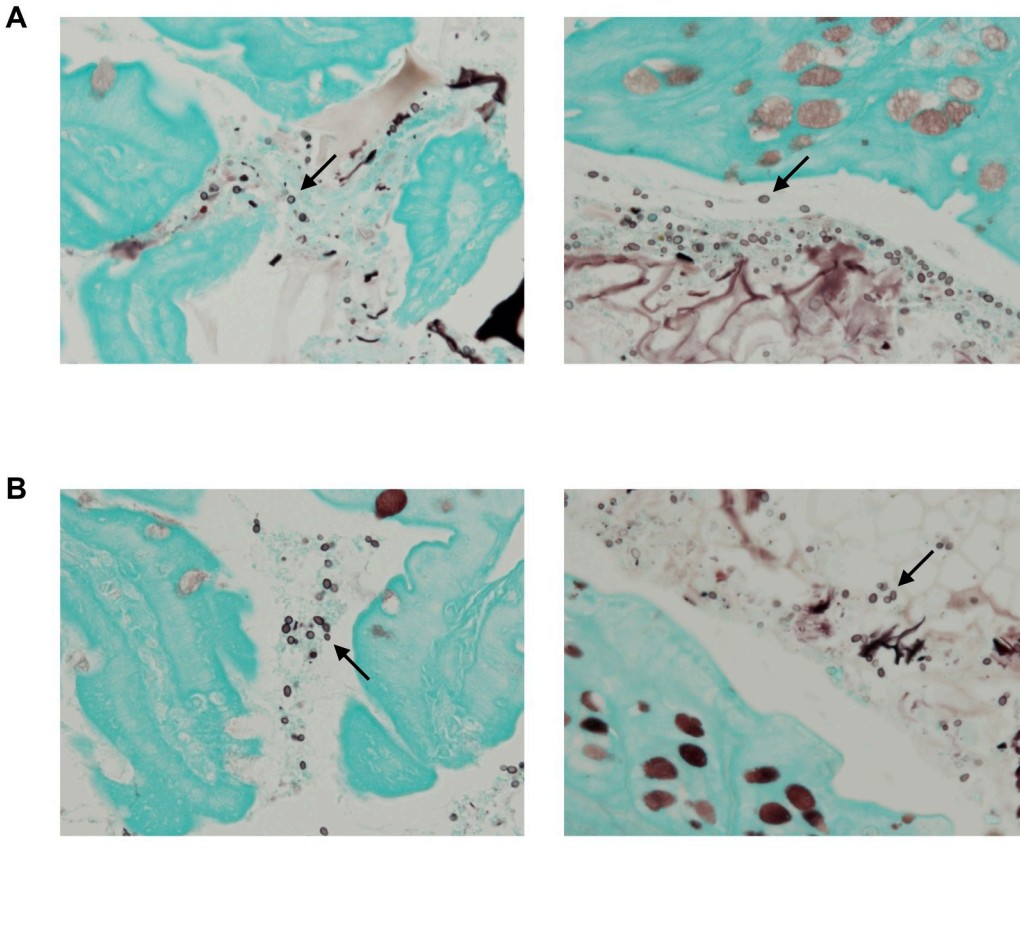

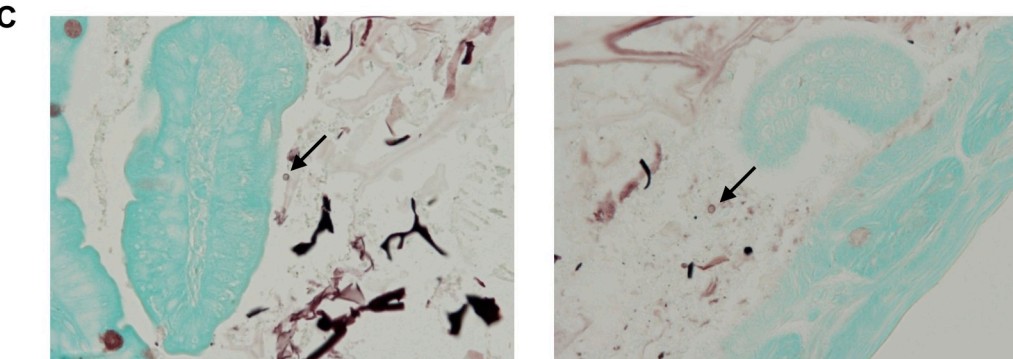

**Fig 2. The invasive strains of *Candida auris* demonstrated greater colonization in murine intestines than the non-invasive strains.** (A–C) GMS staining of the colon 14 days after inoculation with the invasive strain AR0382 (A), the invasive strain AR0390 (B), and the non-invasive strain JCM 15448 (C). Arrows indicate yeast cells of *C. auris*.

disseminating from the gastrointestinal tracts under cortisone acetate immunosuppression than the non-invasive strain isolated from the external ear canal.

## Biofilm-forming capability of *C. auris*

Given the results of colonization, dissemination, and histopathological analysis, we evaluated the biofilm-forming capability of each *C. auris* strain. In this analysis, the invasive strain of *C.*

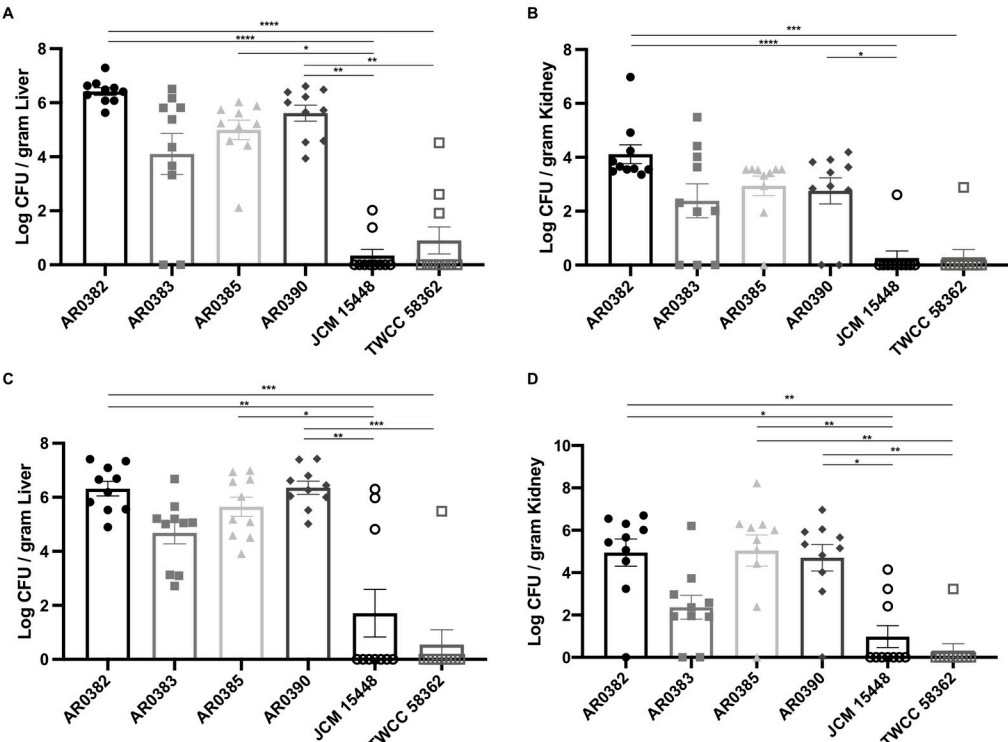

**Fig 3. The invasive strains of *Candida auris* demonstrated greater dissemination into organs than the non-invasive strains.** (A–D) The fungal burdens in the liver and kidneys 7 days (A, B) or 14 days (C, D) after inoculation with invasive or non-invasive *C. auris* strains. Liver and kidney fungal burdens are shown as the Log CFU/g liver and Log CFU/g kidney, respectively. All results are expressed as mean ± standard error of the mean from six independent experiments with a total of 10 tissue samples per group. $^{*}$ $P < 0.05$, $^{***}$ $P < 0.001$, $^{****}$ $P < 0.0001$.

*auris*, AR0382, exhibited the highest levels of biofilm formation, and higher than the non-invasive *C. auris* strains (Fig 5). In addition, the other invasive strains were able to produce biofilms more effectively than the non-invasive strains. On the other hands, the non-invasive strain, especially JCM 15448, produced few biofilms. These results indicated that the invasive *C. auris* strains presented higher biofilm-forming capabilities, which may be associated with greater ability of colonizing the gastrointestinal tracts and disseminating to the organs.

## Survival analysis following dissemination of endogenous *Candida* species

Given the results of the organ CFU counts and histopathological analysis, we chose to use the invasive strain AR0382 and the non-invasive strain JCM 15448 in the survival analysis. In addition, we used *C. albicans* SC5314 and *C. glabrata* CBS 138 as comparisons. In this experiment, we observed and compared the survival rate and body weight changes after inoculation with each *Candida* species. All of the mice which reached the endpoint criteria were euthanized and no mice died before meeting criteria for euthanasia. The results demonstrated that the mortality rate was significantly higher in mice infected with *C. albicans* than those inoculated with other *Candida* species ($P < 0.0001$, Fig 6A). The weight loss of mice infected with *C. auris* AR0382 was significantly higher than that of those infected with *C. auris* JCM 15448 or *C. glabrata* (Day 15; $P < 0.05$, Day 17; $P < 0.001$, Days 19 and 21; $P < 0.0001$ Fig 6B). However, no significant differences were noted in the survival rates among groups other than those infected with *C. albicans*, despite the deaths of several mice following *C. auris* inoculation.

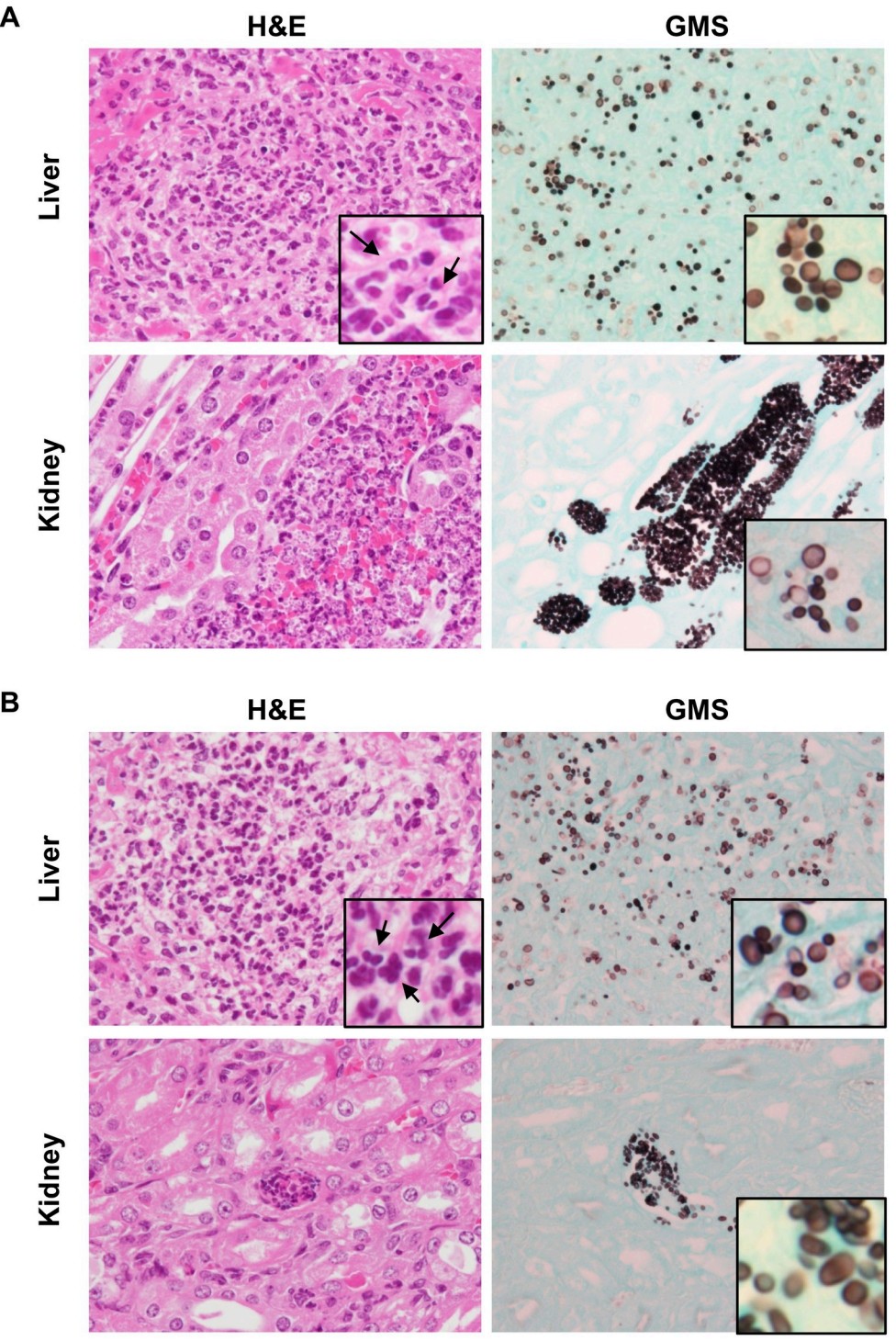

**Fig 4. The invasive strains of *Candida auris* showed massive fungal dissemination.** (A-B) Histopathological analysis of the liver and kidney of mice infected with invasive strains AR0382 (A) and AR0390 (B), 14 days after inoculation. Hematoxylin and eosin staining (left panels) and GMS staining (right panels) are shown. Original magnification: ×400 (inset: ×1000). Neutrophils are indicated by arrows (insets).

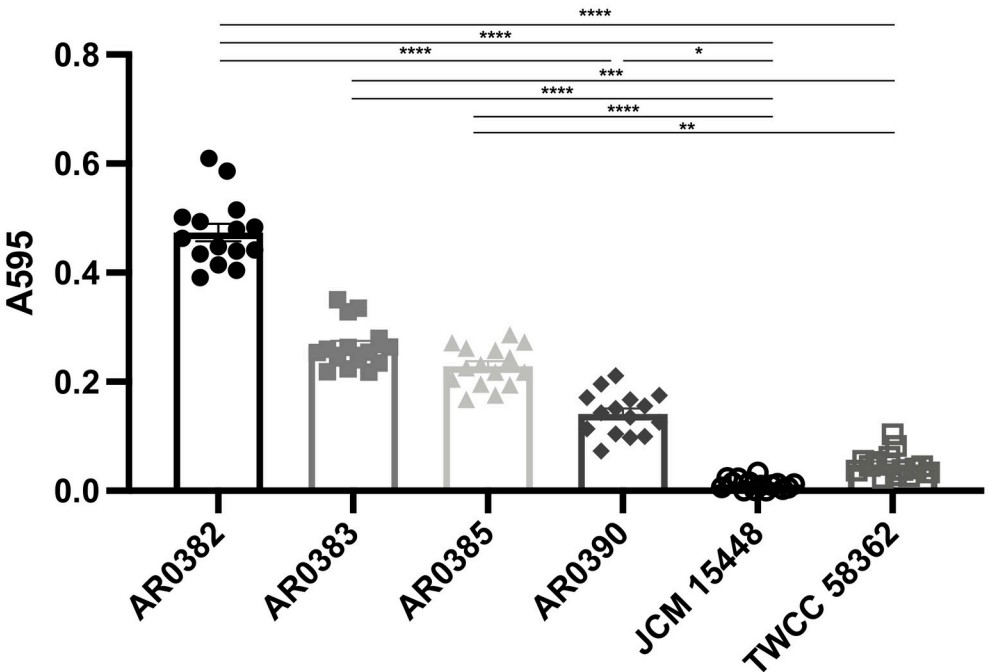

**Fig 5. The invasive strains of *Candida auris* showed higher biofilm-forming capacity than the non-invasive strains.** The biofilm values of each *C. auris* strain stained with crystal violet are shown. Biofilm values are given as optical density at 595nm. All results are expressed as mean ± standard error of the mean from three independent experiments with a total of 15 samples per group. ** $P < 0.01$, **** $P < 0.0001$.

Collectively, the invasive *C. auris* isolates were able to colonize and disseminate more effectively than the non-invasive *C. auris* isolates, leading to the higher virulence of invasive strains from the viewpoint of body weight changes. However, according to our endogenous *Candida* species dissemination mouse model, the mortality attributed to *C. auris* was lower than that for *C. albicans*, and no significant differences were noted among the strains of *C. auris*.

## Discussion

*Candida auris* has recently been reported worldwide, and is believed to be a threat due to the potency of its multidrug-resistance and high virulence [11,15,16]. Globally, *C. auris* has been primarily isolated from patients with bloodstream infections, although all strains in Japan belonging to the East Asian clade were isolated from the external ear canals of infected patients. Therefore, it was hypothesized that there are differences in the virulence and colonization capabilities between the non-invasive isolates from Japan and the invasive strains isolated in other countries. In this study, we evaluated the capability of the colonization and dissemination of each *C. auris* strain using an endogenous dissemination mice model under treatment with cortisone acetate. The results demonstrated that the invasive strains, especially the South Asian clade, of *C. auris* were more effective at colonization and dissemination from the gastrointestinal tract than the non-invasive strains obtained from external ear canals.

During the analysis of fungal burden in organs and stools, both the rates of colonization in the gastrointestinal tract and dissemination to organs were significantly higher in mice infected with the invasive strains compared with those inoculated with the non-invasive strains. In addition to these results, the biofilm-forming capabilities of the invasive strains were shown to be higher than those of the non-invasive strain. Reports have indicated that the

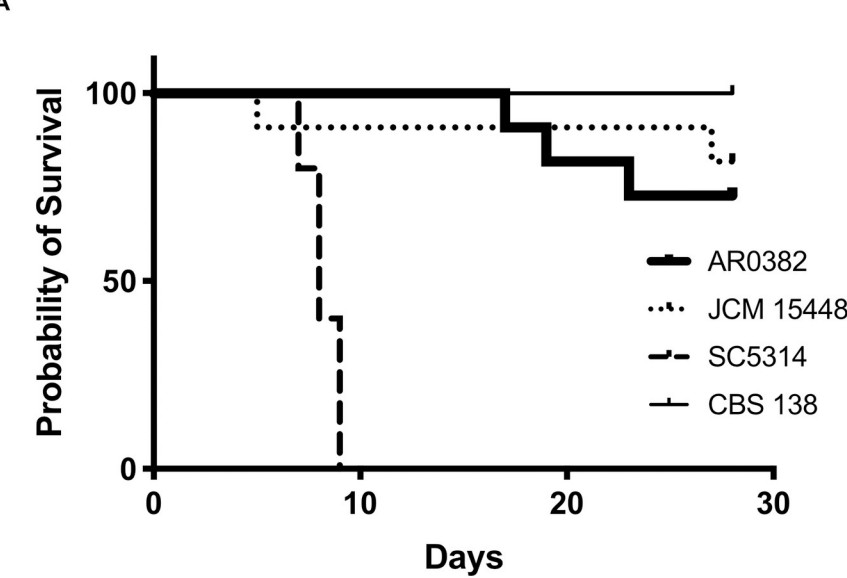

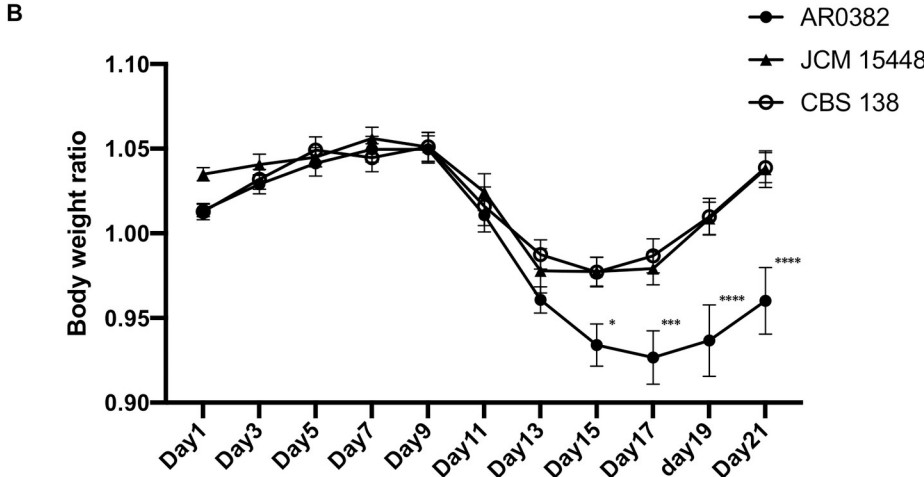

**Fig 6. The invasive strains of *Candida auris* resulted in greater weight loss than the non-invasive strain and *Candida glabrata* despite of lower mortality rate compared with *Candida albicans*.** (A, B) The survival rates (A) and body weight changes (B) of mice inoculated with the invasive strain of *C. auris* AR0382, the non-invasive strain JCM 15448, the reference strain of *C. albicans* SC5314, and the reference strain of *C. glabrata* CBS 138 are shown. Body weight change results are expressed as mean ± standard error of the mean from three independent experiments including 16 *C. auris*–infected mice and 10 *C. albicans* of *C. glabrata*–infected mice. * $P < 0.05$, *** $P < 0.001$, **** $P < 0.0001$.

biofilms of *C. albicans* are associated with colonization on mucosal surfaces such as the oral cavity and vagina [28,29]. In addition, the biofilm is associated with resistance to the immune system and antifungal drugs [30–32]. Indeed, no reports have yet described the relationship between biofilm formation in *C. auris* and its potency in terms of colonization or virulence, although clinical data have suggested that biofilm formation is associated with *C. albicans* pathogenicity and patient deaths [33]. In our study, the highly effective formation of a biofilm by *C. auris* appeared to be associated with the capability to colonize the gastrointestinal tract, although further investigation of the direct relationship between biofilm formation and

pathogenicity of *C. auris* is warranted. In addition, our investigation showed that *C. auris* isolates belonging to the same clade had different biofilm-forming capacities, however, there are no reports describing these differences. Further investigation is also necessary to clarify these differences.

Histopathological analyses revealed that the round cell form of *C. auris* was disseminated under cortisone acetate immunosuppression. In our experiments, the invasive strains of *C. auris* disseminated into organs including the liver and kidneys without establishing a filamentous form, which is accordant with previous reports [9,17,18]. A previous study demonstrated the ability of *C. auris* to generate a filamentous form under limited conditions, and this filamentous phenotype is associated with its invasive capacity [34]. However, our results did not demonstrate that *C. auris* had a filamentous phenotype in spite of confirmed colonization in colons and dissemination into organs, which implies that forming a filamentous phenotype is not the only factor associated with pathogenicity. In addition, the accumulation of neutrophils around the yeasts was histopathologically determined, which indicated that neutrophils play a role in protecting against *C. auris* invasion under cortisone acetate immunosuppression. However, a previous report revealed that *C. auris* can evade neutrophil attacks [35]. In addition, clinical investigations did not find that neutropenia was a common risk factor for infection of the bloodstream by *C. auris* [16,36]. Taken together, although neutrophils may be involved in defense against *C. auris* infection under certain conditions, it remains to be elucidated whether or not neutrophils are generally important factors in protecting patients against *C. auris* infection. Further investigation will be necessary for the clarification of the immunological response to systemic *C. auris* infection.

The survival analysis established that *C. albicans* is more pathogenic than *C. auris*. In our analysis, several mice infected by *C. auris* AR0382 died; however, the mortality rates of these mice were generally low. A number of previous reports have indicated a high rate of mortality following *C. auris* infection in experiments involving mice [17–19]. The reasons for the discrepancies between previous reports and our study may be due to differences in the mice strains used in the experiments, variances in the route of infection, and the type of immunosuppression used. As susceptibility to systemic *C. auris* infection differs between mice strains, we must note that strain C57BL/6J was used in the experiments reported in the present study, although ICR, BALB/c, or A/J strains were used in the studies reported elsewhere [17–19]. In addition, C57BL/6J mice have been reported to be more resistant to *C. auris* infection than A/J mice [19]. The differences in the infection route could be a more attributable point. Intravenous inoculation is considered to cause a more severe infection than other routes of exposure. Our experiments imitated endogenous dissemination from the gastrointestinal tract; therefore, the severity of the infection may have been lower than that observed in an intravenous infection model, leading to the lower rates of mortality noted in this study. The final point of difference was the type of immunosuppression implemented. Cortisone acetate was used in our experiments, although cyclophosphamide was used in most of the previous reports [18,19]. Cyclophosphamide can cause severe neutropenia, leading to high mortality rates. Conversely, cortisone acetate is considered to suppress cellular immunity but not cause neutropenia. The accumulation of neutrophils was noted in organs during our experiments, which implies that immune responses to *C. auris* remained, although their aptitude for phagocytosis or killing capability may have been suppressed.

There were some limitations encountered in our experiments. First, gastrointestinal colonization of *C. auris* was relatively rare compared with skin colonization because of poor growth under anaerobic conditions, therefore, the rate of endogenous candidiasis of *C. auris* was thought to be small [37]. However, some previous reports have described the gastrointestinal colonization of *C. auris*, and our investigations showed the possibilities of *C. auris*

gastrointestinal colonization and dissemination under corticosteroid immunosuppression [21,22]. In addition, *C. auris* isolates could grow on agars under anaerobic conditions, although growth of *C. auris* were poor compared with *C. albicans* or *C. glabrata* (S2 Fig). These mouse models are not necessarily relevant to human events, however, the facts that *C. auris* could colonize and disseminate from gastrointestinal tracts under immunosuppressive conditions are thought to be significant. Based on our investigations, clinicians should pay attention to *C. auris* colonization and dissemination, especially in immunocompromised hosts. Second, we describe the differences in colonization ability and dissemination of non-invasive and invasive strains, although the detailed factors associated with these differences remain unknown. Previous reports have established that several genes and enzymes are associated with virulence in *C. auris* [37,38]; however, we did not investigate the differences of these virulence factors in this experiment and further investigations in this area are, therefore, warranted.

In summary, to the best of our knowledge, this is the first report to describe the gastrointestinal colonization and dissemination into organs of different strains of *C. auris* in a mouse model using cortisone acetate-induced immunosuppression. The invasive strains demonstrated greater capability for both colonization and dissemination from gastrointestinal tracts than the non-invasive strain in this model. The effect on mortality of invasive *C. auris* strains was lower than that of *C. albicans*; however, the capability for endogenous dissemination of *C. auris* from gastrointestinal tracts was thought to be a major factor in the invasive nature of this microorganism, especially in strains exhibiting multidrug resistance.

## Supporting information

**S1 Fig. The non-invasive strain of *Candida auris* showed no evidence of dissemination.** Histopathological analysis of the liver and kidney of mice infected with non-invasive strain JCM 15448, 14 days after inoculation. Hematoxylin and eosin staining (left panels) and GMS staining (right panels) are shown. Original magnification: ×200.
(TIF)

**S2 Fig. The growth of *Candida albicans*, *Candida glabrata*, and each strain of *Candida auris* on YPD agar under anaerobic condition.** Anaerobic growth of *Candida albicans* (SC5314), *Candida glabrata* (CBS 138), and each strain of *Candida auris* on YPD agar, 7 days after inoculation is shown.
(TIF)

## Author Contributions

**Conceptualization:** Masahiro Abe, Yoshitsugu Miyazaki.

**Data curation:** Masahiro Abe.

**Formal analysis:** Masahiro Abe.

**Funding acquisition:** Harutaka Katano, Hideki Hasegawa, Yoshitsugu Miyazaki.

**Investigation:** Masahiro Abe, Harutaka Katano, Yuko Sato.

**Methodology:** Masahiro Abe, Harutaka Katano.

**Project administration:** Yoshitsugu Miyazaki.

**Resources:** Ken Kikuchi, Hideki Hasegawa, Yoshitsugu Miyazaki.

**Supervision:** Yoshitsugu Miyazaki.

**Visualization:** Harutaka Katano.

**Writing – original draft:** Masahiro Abe.

**Writing – review & editing:** Harutaka Katano, Minoru Nagi, Yoshitsugu Higashi, Yuko Sato, Ken Kikuchi, Hideki Hasegawa, Yoshitsugu Miyazaki.

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
