## [Decision Letter · Decision Letter 0]

24 Sep 2020

PONE-D-20-21345

Potency of gastrointestinal colonization and virulence of *Candida auris* in a murine endogenous candidiasis

PLOS ONE

Dear Dr. Miyazaki,

Thank you for submitting your manuscript to PLOS ONE. After careful consideration, we feel that it has merit but does not fully meet PLOS ONE’s publication criteria as it currently stands. Therefore, we invite you to submit a revised version of the manuscript that addresses the points raised during the review process.

We look forward to receiving your revised manuscript.

Kind regards,

Shankar Thangamani, DVM, PhD

Academic Editor

PLOS ONE

3.PLOS requires an ORCID iD for the corresponding author in Editorial Manager on papers submitted after December 6th, 2016. Please ensure that you have an ORCID iD and that it is validated in Editorial Manager. To do this, go to ‘Update my Information’ (in the upper left-hand corner of the main menu), and click on the Fetch/Validate link next to the ORCID field. This will take you to the ORCID site and allow you to create a new iD or authenticate a pre-existing iD in Editorial Manager. Please see the following video for instructions on linking an ORCID iD to your Editorial Manager account: https://www.youtube.com/watch?v=_xcclfuvtxQ

4. Please ensure that you refer to Figure 4 in your text as, if accepted, production will need this reference to link the reader to the figure.

<h1>** **</h1>

Reviewers' comments:

Reviewer's Responses to Questions

**Comments to the Author**

1. Is the manuscript technically sound, and do the data support the conclusions?

Reviewer #1: Yes

2. Has the statistical analysis been performed appropriately and rigorously? 

Reviewer #1: Yes

3. Have the authors made all data underlying the findings in their manuscript fully available?

Reviewer #1: Yes

4. Is the manuscript presented in an intelligible fashion and written in standard English?

Reviewer #1: No

5. Review Comments to the Author

Reviewer #1: Overall the paper is clearly presented and the conclusions appear sound based on the available evidence. Minor grammatical corrections are needed, and some methodological clarifications are detailed below.

• Line 54: The authors mention exogenous infection here but doesn’t elaborate on any clinical significance associated with this form of infection.

• Line 56: An article is needed between “particular” and “gastrointestinal”

• Line 125-126: Why were kidney homogenates not plated on YPD agar with antibiotics? What is the rationale behind the different antibiotic conditions for these homogenates?

• Line 144: Grammatically this line is difficult to read, specifically “removed for excluding”.

• Line 158: What physiological signs were used to designate a mouse as moribund?

• Line 241-243: Is there anything about AR0390 that suggests why it produced biofilms at a substantially lower level than AR0382?

• Line 259: Grammatically change “All of mice” to “All of the mice”.

6. PLOS authors have the option to publish the peer review history of their article (what does this mean?). If published, this will include your full peer review and any attached files.

Reviewer #1: No

---

## [Author Response · Author response to Decision Letter 0]

28 Oct 2020

Thank you very much for your helpful suggestions and comments to our manuscript. We assessed each comment one by one.

---

## [Editor Report · Decision Letter 1]

18 Nov 2020

Potency of gastrointestinal colonization and virulence of *Candida auris* in a murine endogenous candidiasis

PONE-D-20-21345R1

Dear Dr. Miyazaki,

We’re pleased to inform you that your manuscript has been judged scientifically suitable for publication and will be formally accepted for publication once it meets all outstanding technical requirements.

Kind regards,

Shankar Thangamani, DVM, PhD

Academic Editor

PLOS ONE

---

## [Editor Report · Acceptance letter]

20 Nov 2020

PONE-D-20-21345R1 

Potency of gastrointestinal colonization and virulence of *Candida auris* in a murine endogenous candidiasis 

Dear Dr. Miyazaki:

I'm pleased to inform you that your manuscript has been deemed suitable for publication in PLOS ONE. Congratulations! Your manuscript is now with our production department. 

Kind regards, 

on behalf of

Dr. Shankar Thangamani 

Academic Editor

PLOS ONE